# Accuracy and reproducibility of tumor size measurement using a deep-learning–based CDSS in resected lung cancer

**Eun Young Kim[1,2], Jun Seong Kim[2], Kwang Nam Jin[3,4], Young Jun Cho[5,6], Jong-Yeup Kim** [ORCID][7,8]*

**1** Radiology Department, Incheon Sejong Hospital, 20, Gyeyangmunhwa-Ro, Gyeyang-Gu, Incheon, Republic of Korea, **2** Department of Radiology, Gil Medical Center, Gachon University College of Medicine, Incheon, South Korea, **3** Department of Radiology, Boramae Medical Center, Seoul, South Korea, **4** Seoul National University College of Medicine, Seoul, South Korea, **5** Department of Radiology, Konyang University Hospital, Daejeon, Korea, **6** Konyang University School of Medicine, Daejeon, Korea, **7** Department of Biomedical Informatics, Konyang University College of Medicine, Daejeon, Republic of Korea, **8** Department of Otorhinolaryngology-Head and Neck Surgery, Konyang University College of Medicine, Daejeon, Republic of Korea

* jykim@kyuh.ac.kr

## Abstract

### Purpose

MONCAD LCT is a commercially available deep-learning based clinical decision support system (CDSS) for lung screening CT. The aim of this multicenter retrospective study was to evaluate the accuracy and reproducibility of tumor size measurement using a commercially available deep-learning–based clinical decision support system (CDSS), compared with radiologist assessments and pathological reference in resected lung cancer cases.

### Methods and materials

We retrospectively collected preoperative CT images and original radiology reports and the CDSS results for resected lung cancer from three institutions during 2022 (n = 176). MONCAD LCT evaluated the LungRADs category based on the density and size of the lung nodule. First of all, we compared the MONCAD LCT and original radiologic report using the pathologic tumor size as gold standard. Furthermore, the subsampling case (n = 33) randomly selected by institutions, density type (pure ground glass opacity, subsolid, and solid) and tumor size, two expert thoracic radiologists independently evaluated the tumor size for the resected lung cancer and the interobserver variability was evaluated.

### Results

Among 176 tumors, 162 (92%) were detected on MONCAD LCT. Tumor size measurement by original radiology report and CDSS were found to have excellent

**Data availability statement:** All relevant data are within the manuscript and its Supporting Information files.

**Funding:** This research was supported by a grant of the Korean Health Technology R&D Project through the Korea Health Industry Development Institute (KHIDI), funded by the Ministry of Health & Welfare, Republic of Korea (grant number: RS-2022-KH129742). The funders had no role in study design, data collection and analysis, decision to publish, or preparation of the manuscript.

**Competing interests:** none.

reliability with pathologic tumor size (ICC = 0.869 for absolute agreement). On reader study, excellent interobserver agreement (ICC = 0.907) was observed between two expert radiologists, which was inferior to the completely consistent CDSS results (ICC = 1.000).

## Conclusions

No significant differences were found in the measurement of tumor size between radiologists and the CDSS. CDSS might be helpful to minimize interobserver variability for tumor size measurement by supplying consistent and reliable results.

## *Clinical Relevance/Application

This real-world multicenter study demonstrates that the CDSS provides consistent and objective tumor size measurements, supporting its potential utility in standardizing preoperative lung cancer assessment.

## Introduction

Lung cancer is the second most commonly diagnosed cancer in both men and women, with non-small-cell lung cancer (NSCLC) comprising 85% of cases [1]. Accurate measurement of tumor size is a critical component in the staging of NSCLC, as it directly influences the T descriptor in the TNM classification and subsequently guides treatment decisions and prognostication [2]. Chest computed tomography (CT) remains the primary imaging modality for preoperative tumor evaluation, yet interobserver variability in size measurement remains a challenge, especially in cases with complex morphology or subsolid components.

Recent advances in artificial intelligence (AI) and computer-aided detection (CAD) systems have introduced automated tools capable of analyzing pulmonary nodules with high consistency [3–8]. Several prior studies have demonstrated that AI-based systems can achieve comparable or even superior reproducibility in nodule size measurements compared to human readers. However, most of these studies have been limited to screening populations or focused on benign or indeterminate nodules, without pathologic confirmation. Additionally, many have been conducted in single-center settings with relatively homogeneous datasets, limiting their generalizability.

In this multicenter retrospective study, we aimed to evaluate the performance of a commercially available clinical decision support system (CDSS), MONCAD LCT, in measuring tumor size in pathologically confirmed NSCLC cases. We compared its measurements to both manual radiologist reports and the pathologic gold standard, while also assessing detection performance and inter-observer variability. By including a diverse dataset from three tertiary institutions and incorporating a direct comparison to surgical pathology, our study addresses a gap in the literature and offers clinically relevant insights into the utility and limitations of AI-based measurement tools in lung cancer staging.

## Materials and methods

This retrospective cohort study was approved by the institutional review boards of three participating institutions (approval number: GFIRB2023−192 for Gil Medical Center, 10-2023-39 for Boramae Medical Center, 2023-06-009-004 for Konyang University Hospital). The database was reviewed for data collection between January 1 and December 31, 2022, and data access commenced simultaneously at all study sites on August 1, 2023. All data were fully anonymized before you accessed them and the requirement for written informed consent was waived. None of the authors have any financial interests or conflicts of interest with the industry or the product used in this study. The authors maintained full control of the data.

### Study population

We retrospectively identified consecutive patients who underwent surgery for NSCLC between January and December 2022 at three institutions. Eligible patients had undergone preoperative chest CT, and cases were retrieved from the radiology database and electronic medical records. Patients were excluded if the interval between the preoperative CT and surgery exceeded one month, if they had multifocal tumors requiring resection of multiple lesions, if the CT images contained severe artifacts that impaired evaluation, or if tumor size could not be measured due to an endobronchial location or severe post-obstructive pneumonia. Finally, a total of 176 tumors were included in the final analysis. Data on patient age, sex, primary tumor location, exam dates of chest CT, and corresponding radiology reports were retrospectively collected. Resected lung cancers were evaluated for location, density type and size using the original radiology reports.

For tumor size assessment, the maximum diameter documented in the radiology report was extracted. In cases of part-solid nodules, when both total and solid-component diameters were documented, the diameter of the solid component was consistently used for analysis, in accordance with the 8th edition of the TNM classification by the IASLC, which recommends solid component measurement for T-staging. If only one measurement was present, that value was used as reported. All values were directly extracted from the original clinical reports without reinterpretation or remeasurement.

All chest CT scans were acquired using multi-detector CT scanners with at least 64 detector rows across the three participating institutions (Gachon University Gil Medical Center, Konyang University Hospital, and Boramae Medical Center using standard clinical protocols for thoracic imaging (S1 Table).

### Deep-learning based CDSS on preoperative CT

MONCAD LCT (MONCAD LCT for Chest CT Version 1.00.04; Monitor Corporation Co., Ltd., Seoul, South Korea) is a commercially available CDSS designed to assist in the interpretation of chest CT scans. MONCAD LCT is the international version of the AI-CAD system known as LuCAS-plus in South Korea. Both refer to the same product developed by Monitor Corporation.

To briefly describe the system architecture, MONCAD LCT is based on a 3D convolutional neural network trained on the LIDC-IDRI dataset, which includes expert-labeled CT scans with pulmonary nodules [9] (S2 table). The system performs automated detection and segmentation of nodules, and the size is determined using the largest 2D measurement on the axial plane. For part-solid nodules, the solid portion is defined as the region with CT attenuation greater than –350 Hounsfield Units (HU), with low-density areas excluded prior to measurement. It automatically analyzes pulmonary nodules, providing measurements of tumor size, density classification (solid, part-solid, non-solid), location, and Lung-RADS categorization.

Preoperative CT scans were retrospectively reanalyzed using the commercially available CDSS, MONCAD LCT, which enabled automated extraction of tumor-related features, including size, density type, and other diagnostic parameters. Tumor size was defined as the maximum diameter of the solid component of the nodule and was used for all measurements and analyses (Fig 1).

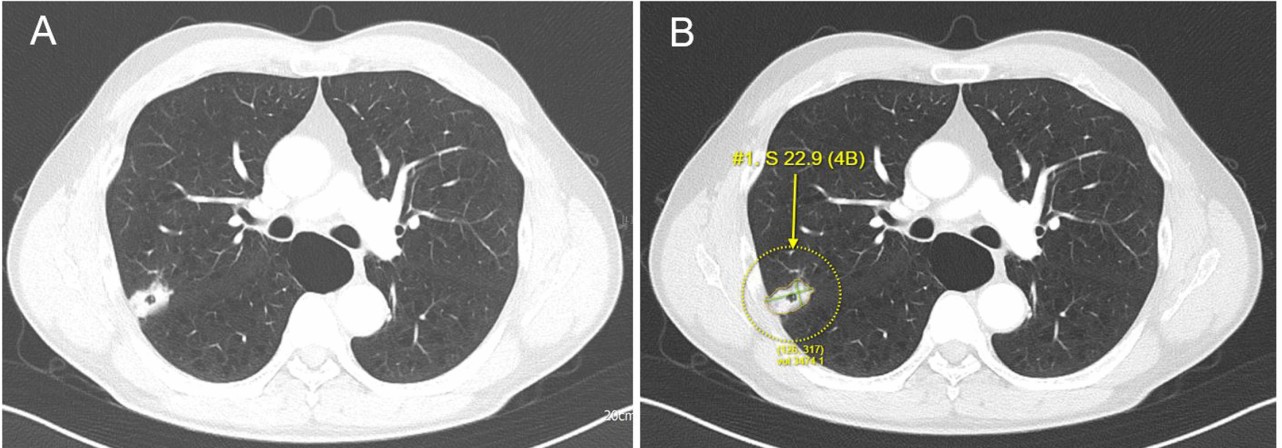

**Fig 1. A representative case.** A 67-year-old male who underwent right upper lobectomy for lung adenocarcinoma. On pathology, the tumor size was 28 mm (T1c). The maximal tumor size was 27 mm on original radiology report on preoperative CT scan (A) and the MONCAD LCT result (B) shows the maximal tumor size as 29 mm. This figure was generated from de-identified clinical CT data and edited by the authors. Reprinted under CC BY license with author permission.

### Reader study using for Inter-Observer Variability

A subset of 33 cases was selected using stratified random sampling based on contributing institution, tumor density type (solid, part-solid, or pure ground-glass nodule), and tumor size (<or ≥, mean tumor size of 21 mm). Two board-certified thoracic radiologists (TR1 and TR2 with 21, and 15 years of experience in thoracic imaging, respectively) independently reviewed the selected cases and assessed each tumor for density classification and maximum diameter. Inter-observer variability between TR1 and TR2 was then evaluated to determine consistency in tumor size measurement.

### Ground truth based on pathology reports for resected lung cancer

All references were based on the pathologic tumor size and staging. The CT sizes and staging of the primary tumors were estimated by original radiology report or by the CDSS and compared as a function of pathologic sizes and staging. The maximal macroscopic dimensions of the primary lesion were documented by the pathologists. Of these three-dimensional diameters, the largest one was considered to be the reference tumor size.

### Statistical analysis

The results are presented as percentages for categorical variables and as means (± standard deviation) for continuous variables.

For the entire cohort, Pearson correlation analysis was performed to assess the relationship between tumor size measurements from the original radiologic reports and the MONCAD LCT system, using the pathological tumor size as the reference standard. In addition, intraclass correlation coefficients (ICCs) were calculated to evaluate the absolute agreement among the original radiologic reports, MONCAD LCT measurements, and pathologic findings.

For the reader study conducted on the subsampled cases (n = 33), inter-observer variability between the two thoracic radiologists was assessed to determine consistency in tumor size measurement.

Statistical analyses were performed using MedCalc version 19.5.1 or R version 3.5.3. *P*-values less than 0.05 were considered to indicate significant differences.

# Results

## Baseline characteristics of the study population

Table 1 shows the demographic features of the study subjects (101 males and 74 females; mean age, 66±9 years). The average tumor size was 21 mm. Regarding density type, 101 tumors (57.4%) were classified as solid, 54 (30.7%) as part-solid, and 21 (11.9%) as pure ground-glass nodules (GGNs). Most tumors were peripherally located (76.1%), while 23.9% were centrally located. Histologically, adenocarcinoma was the most common subtype, accounting for 145 cases (82.4%), followed by squamous cell carcinoma (n=26, 14.8%) and other histologic types (n=5, 2.8%).

## Deep-learning based CDSS on preoperative CT

Among the 176 tumors included in the study, 162 tumors (92%) were successfully detected by the MONCAD LCT system. The remaining 14 tumors (8%) were not identified by the system. Failure to detect tumors was most commonly observed in small ground-glass nodules (GGNs), where lesion conspicuity on CT might be limited. In addition, detection errors were

**Table 1. Demographic description of the dataset.**

| Characteristics | All tumors |
|---|---|
| No. of tumors | 176 |
| Gil medical center | 82 (46.6) |
| Konyang university hospital | 48 (27.3) |
| Boramae medial center | 46 (26.1) |
| Age of patients (years) | 66±9 |
| Sex of patients (men/women) | 101/74 |
| Tumor size (mean size=21 mm) | |
| < 21 mm | 115 (65.3) |
| ≥ 21 mm | 61 (34.7) |
| Density type | |
| Solid | 101 (57.4) |
| Part-solid | 54 (30.7) |
| Pure GGN | 21 (11.9) |
| Tumor location | |
| Lobe | |
| RUL | 69 (39.2) |
| RML | 11 (6.3) |
| RLL | 33 (18.8) |
| LUL | 43 (24.4) |
| LLL | 20 (11.4) |
| Central/peripheral | |
| Central | 42 (23.9) |
| Peripheral | 134 (76.1) |
| Histology | |
| Adenocarcinoma | 145 (82.4) |
| Squamous cell carcinoma | 26 (14.8) |
| Others | 5 (2.8) |

Note: Except where indicated, data are the mean (± SD) or number (%). SD=standard deviation

observed even in some tumors that were sufficiently large to be identified, suggesting that factors beyond size—such as complex morphology or atypical location—may have contributed to the failure. In particular, several of these undetected tumors were abutting or involving the chest wall, which may have hindered accurate recognition by the automated system (Fig 2).

A strong correlation was observed between the tumor size recorded in the original radiologic reports and the pathological tumor size (R = 0.738, R² = 0.544, P < 0.001). Similarly, the correlation between the MONCAD LCT–measured tumor size and the pathological size was also high (R = 0.668, R² = 0.446, P < 0.001) (Fig 3).

Intraclass correlation coefficient (ICC) analysis further confirmed excellent agreement between both radiologic and MONCAD LCT –derived tumor measurements and the pathological reference. The ICC for absolute agreement was 0.869 (95% CI: 0.814–0.907, P < 0.001), indicating a high level of reliability in size measurement across all modalities.

In the reader study, excellent inter-observer agreement was observed between the two expert thoracic radiologists, with an ICC of 0.907 (95% CI: 0.689–0.963, P < 0.001). While this indicates a high level of consistency, it was lower than the perfectly consistent measurements obtained from the MONCAD LCT system (ICC = 1.000), reflecting the fully automated and reproducible nature of the AI-based analysis.

## Discussion

This study evaluated the measurement accuracy and consistency of a commercially available CDSS for preoperative CT-based tumor size assessment, using a multicenter retrospective cohort of patients with resected non-small cell lung cancer (NSCLC). The MONCAD LCT successfully detected 92% of resected lung tumors. Tumor size measurements showed strong correlation and excellent agreement with pathological reference standards, both for radiologist assessments and the automated outputs. Inter-observer variability between expert radiologists was high, but slightly lower than the perfectly consistent measurements provided by MONCAD LCT, highlighting the potential utility of CDSS in standardizing tumor evaluation.

Several previous studies have explored the role of AI and CAD in the assessment of lung tumor size on chest CT [3–8]. These tools have shown promising accuracy and reproducibility, particularly in measuring nodule diameter and volume. For example, prior work has demonstrated that deep learning–based algorithms can achieve high concordance

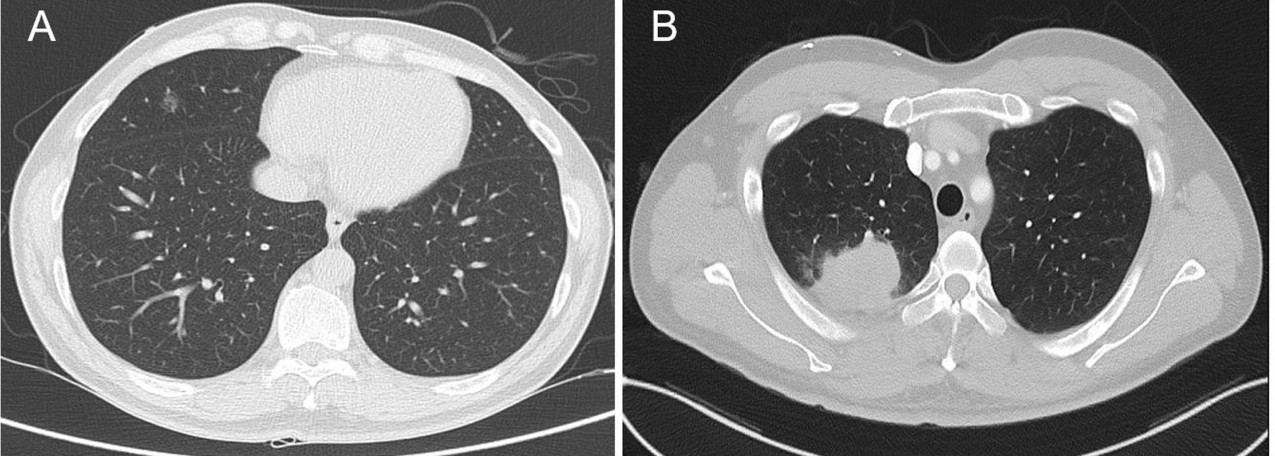

**Fig 2. Examples of detection errors. (A)** A small pure ground-glass nodule (GGN) that was not identified by the system. **(B)** A lesion of sufficient size for detection but missed probably due to its abutment to the chest wall. This figure was generated from de-identified clinical CT data and edited by the authors. Reprinted under CC BY license with author permission.

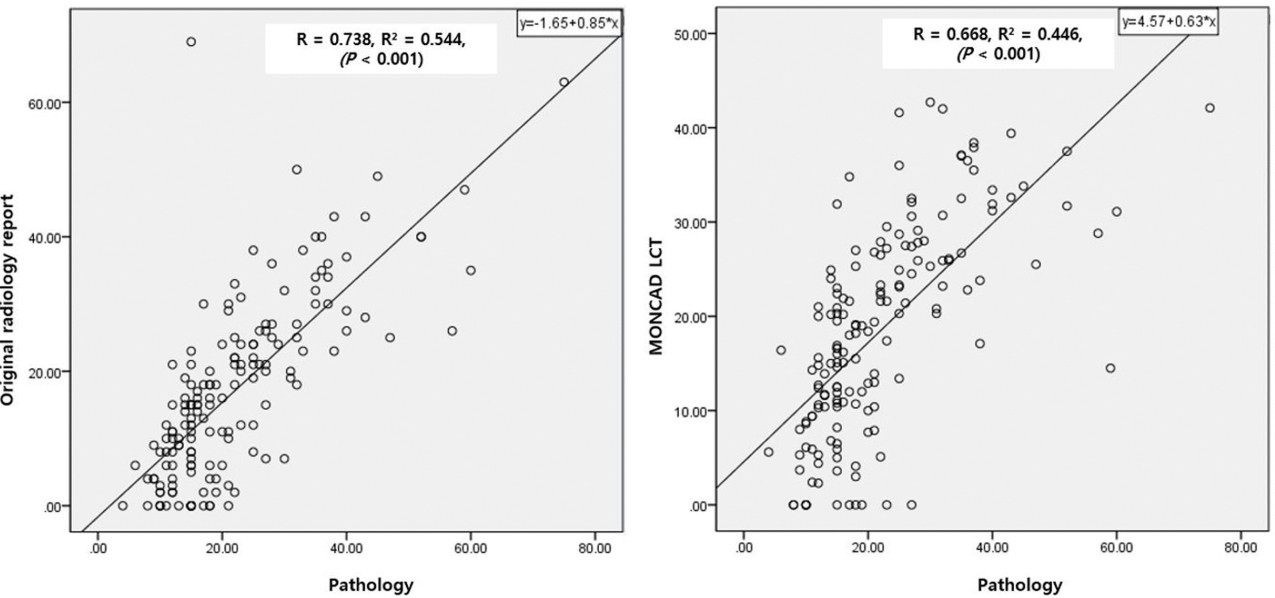

**Fig 3. Scatter plots.** Scatter plots showed strong positive correlation between tumor size measurement of original radiologist assessment (R=0.738, R²=0.544, P<0.001) and automated outputs of MONCAD LCT (R=0.668, R²=0.446, P<0.001) with pathological reference standards.

with radiologist measurements and even outperform human readers in consistency, especially for subsolid and small pulmonary nodules. However, most of these studies have primarily focused on screening populations or incidental nodules, rather than pathologically confirmed lung cancers. Our study contributes to this body of evidence by validating an MFDS-approved, commercially available CDSS (MONCAD LCT) against a pathological reference standard in a surgical cohort. By directly comparing automated measurements with both radiologist assessments and pathologic tumor size, we demonstrated that CDSS can produce reliable and consistent results, which may help reduce inter-observer variability in routine clinical settings.

The failure of MONCAD LCT to detect approximately 8% of resected lung cancers in our cohort may, in part, be attributed to the system's original design and training context. MONCAD LCT was primarily developed as a CDSS for lung cancer screening, with a focus on identifying small pulmonary nodules in asymptomatic populations. As such, its detection algorithms may have been optimized for early-stage, well-circumscribed, and typically peripherally located lesions—commonly encountered in low-dose CT screening protocols. These findings highlight both the potential and the current limitations of commercially available CDSS tools in diverse clinical scenarios. In this study, the role of MONCAD LCT was limited to automated tumor size measurement on preoperative CT, specifically for assessing the solid portion relevant to T-staging in resected NSCLC. Although the system is marketed as a CDSS, we used it in a narrowly defined scope rather than for comprehensive diagnostic support. Clarifying this application is essential to accurately interpret its performance and limitations within the context of our study design.

Despite these shortcomings, deep-learning based CDSS offers several important advantages in the evaluation of pulmonary nodules and lung tumors. First, they provide high reproducibility and consistency, reducing inter- and intra-observer variability that is often seen in manual radiologic assessments. Second, AI systems can process large volumes of imaging data rapidly and objectively, offering efficient support in high-throughput clinical settings, such as lung cancer screening or preoperative staging. Third, advanced algorithms are capable of performing quantitative analyses, such as volumetric measurements and growth rate calculations, which may be more sensitive than traditional diameter-based

methods in detecting early tumor progression. As CDSS tools continue to evolve, further studies are needed to improve their sensitivity in challenging cases and to integrate multimodal data (e.g., imaging, pathology, genomics). Such advancements will be key to expanding their clinical utility in thoracic oncology.

Another key strength of our study is its use of real-world, multicenter data across three institutions, enhancing the generalizability of the findings. While CT scans were acquired using ≥64-channel systems under standardized protocols, minor inter-institutional variations were unavoidable in this retrospective multicenter study. Still, the CDSS yielded reliable results, supporting its applicability across diverse clinical settings. Moreover, by including a reader study with two independent thoracic radiologists, we were able to quantify inter-observer variability and demonstrate that the CDSS provided perfectly consistent size measurements (ICC = 1.000), supporting its potential to reduce variability and offer more objective, reproducible, and accurate tumor size assessments across different readers and institutions.

However, our study also has limitations. The retrospective design may introduce selection bias, and the relatively small sample size—restricted to surgically resected lung cancers—limits applicability to unresected or advanced-stage lesions. In addition, the analysis was confined to surgically resected lung tumors to allow for comparison with pathologic measurements. As such, the performance of the MONCAD LCT system in non-resected or indeterminate pulmonary nodules remains unassessed. Future studies should aim to validate these findings in larger, prospective cohorts that include both resected and unresected tumors. In addition, expanding the evaluation to include a broader spectrum of clinical scenarios—such as screening-detected nodules or lesions with atypical morphology—may help clarify the robustness and clinical applicability of AI-based decision support systems like MONCAD LCT in real-world practice.

## In summary

MONCAD LCT demonstrated strong agreement in tumor size measurements and successfully detected the majority of lung cancers, with only a small proportion (8%) of lesions remaining undetected. The system's high consistency and reproducibility support its potential as a valuable clinical decision support tool. By minimizing inter-observer variability and enabling standardized tumor assessments, AI-based systems like MONCAD LCT may enhance diagnostic accuracy and workflow efficiency in thoracic oncology.

## Supporting information

**S1 Table. Technical Overview of MONCAD LCT (LuCAS-plus).**
(DOCX)

**S2 Table. CT Scanner and Acquisition Parameters at Participating Institutions.**
(DOCX)

## Author contributions

**Conceptualization:** Eun Young Kim, Kwang Nam Jin, Jong-Yeup Kim.

**Data curation:** Eun Young Kim, Young Jun Cho.

**Formal analysis:** Eun Young Kim, Jun Seong Kim, Kwang Nam Jin.

**Funding acquisition:** Jong-Yeup Kim.

**Investigation:** Eun Young Kim.

**Methodology:** Eun Young Kim.

**Project administration:** Jong-Yeup Kim.

**Resources:** Eun Young Kim, Young Jun Cho.

**Software:** Eun Young Kim.

**Supervision:** Eun Young Kim, Kwang Nam Jin, Jong-Yeup Kim.

**Validation:** Eun Young Kim.

**Visualization:** Eun Young Kim.

**Writing – original draft:** Eun Young Kim, Jun Seong Kim, Kwang Nam Jin, Young Jun Cho.

**Writing – review & editing:** Eun Young Kim, Kwang Nam Jin, Young Jun Cho, Jong-Yeup Kim.

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
