## [Decision Letter · Decision Letter 0]

22 Dec 2025

Dear Dr. Kim,

Thank you for submitting your manuscript to PLOS ONE. After careful consideration, we feel that it has merit but does not fully meet PLOS ONE’s publication criteria as it currently stands. Therefore, we invite you to submit a revised version of the manuscript that addresses the points raised during the review process.

We look forward to receiving your revised manuscript.

Kind regards,

Lorenzo Faggioni, M.D., Ph.D.

Academic Editor

PLOS One

Journal Requirements:

3. Please note that PLOS One has specific guidelines on code sharing for submissions in which author-generated code underpins the findings in the manuscript. In these cases, all author-generated code must be made available without restrictions upon publication of the work. Please review our guidelines at https://journals.plos.org/plosone/s/materials-and-software-sharing#loc-sharing-code and ensure that your code is shared in a way that follows best practice and facilitates reproducibility and reuse.

none

6. Please update your submission to use the PLOS LaTeX template. The template and more information on our requirements for LaTeX submissions can be found at http://journals.plos.org/plosone/s/latex.

7. We note that Figure(s) 1 and 2 in your submission contain copyrighted images. All PLOS content is published under the Creative Commons Attribution License (CC BY 4.0), which means that the manuscript, images, and Supporting Information files will be freely available online, and any third party is permitted to access, download, copy, distribute, and use these materials in any way, even commercially, with proper attribution. For more information, see our copyright guidelines: http://journals.plos.org/plosone/s/licenses-and-copyright.

a. You may seek permission from the original copyright holder of Figure(s) 1 and 2 to publish the content specifically under the CC BY 4.0 license.

8. Please remove all personal information, ensure that the data shared are in accordance with participant consent, and re-upload a fully anonymized data set.

9.Please include captions for your Supporting Information files at the end of your manuscript, and update any in-text citations to match accordingly. Please see our Supporting Information guidelines for more information: http://journals.plos.org/plosone/s/supporting-information.

Reviewers' comments:

Reviewer's Responses to Questions

**Comments to the Author**

1. Is the manuscript technically sound, and do the data support the conclusions?

Reviewer #1: No

Reviewer #2: Yes

2. Has the statistical analysis been performed appropriately and rigorously?

Reviewer #1: Yes

Reviewer #2: Yes

3. Have the authors made all data underlying the findings in their manuscript fully available?

Reviewer #1: No

Reviewer #2: Yes

4. Is the manuscript presented in an intelligible fashion and written in standard English?

Reviewer #1: Yes

Reviewer #2: Yes

Reviewer #1: The study addresses an important clinical task; however, several methodological issues need clarification to ensure the soundness of the findings.

1. Title and aim: “predict” vs “measure”.

The manuscript repeatedly refers to “predicting tumour size” and “diagnostic performance”, but the study actually evaluates measurement accuracy and reproducibility of CT-based tumour size compared with pathology and radiologists. Since the reported ICCs show similar agreement between MONCAD LCT and routine reports, the emphasis on “prediction” seems conceptually misleading. I suggest reframing the aim toward measurement accuracy/consistency or defining more precisely what “predict” means in this context.

2. Description of the deep learning–based system.

The Methods section provides very limited information about MONCAD LCT despite describing it as a deep learning–based CDSS. Key functions such as nodule detection are not mentioned in Methods, although detection rates appear in Results. Likewise, the approach to delineating tumour margins, handling part-solid nodules, and determining the maximum solid-component diameter (2D plane vs 3D volumetric search) is not described. Some concise explanation of these components—within the limits of a commercial product—is necessary to understand how the system achieves its reported reproducibility.

3. Extraction and definition of radiology report measurements.

The process by which tumour size values were retrieved from routine radiology reports is insufficiently described, especially for part-solid nodules where both total and solid-component diameters may be documented. Without a clear extraction protocol, it is difficult to interpret whether the “report” measurements used for comparison reflect the original clinical assessment or choices made during retrospective data collection.

4. Use of the term CDSS.

Although MONCAD LCT may be marketed as a CDSS, in this study it supports only a narrow component of NSCLC evaluation (size assessment for T staging). It may be helpful to clarify this limited scope or use a more precise term.

Clarifying these points would significantly strengthen the methodological transparency and interpretation of the work.

Reviewer #2: Well written and interesting. The protocol for acquiring the analysed CT scans is missing, which is known to be one of the factors that can influence the automatic analyses performed by the software. Can you provide?

**Do you want your identity to be public for this peer review?** For information about this choice, including consent withdrawal, please see our Privacy Policy

Reviewer #1: No

Reviewer #2: No

---

## [Author Response · Author response to Decision Letter 1]

2 Feb 2026

18-January-2026

Dear Lorenzo Faggioni, M.D., Ph.D. Academic Editor, PLOS ONE

Thank you very much for the opportunity to revise our original article entitled “Accuracy and Reproducibility of Tumor Size Measurement Using a Deep-Learning–Based CDSS in Resected Lung Cancer (PONE-D-25- 39492).” After carefully reading the reviewer’ and editor’s comments, we have tried to improve the quality and legibility of the manuscript according to the points raised.

In the revised version, changes are indicated by highlighting. Individual points (E-editor’s comments and R-#1 = point 1 made by reviewer) are indicated in red.

PONE-D-25-39492

The diagnostic performance of deep-learning based clinical decision support system (CDSS) to predict tumor size for patients with resected lung cancer.

PLOS One

Dear Dr. Kim,

Thank you for submitting your manuscript to PLOS ONE. After careful consideration, we feel that it has merit but does not fully meet PLOS ONE’s publication criteria as it currently stands. Therefore, we invite you to submit a revised version of the manuscript that addresses the points raised during the review process.

• A letter that responds to each point raised by the academic editor and reviewer(s). You should upload this letter as a separate file labeled 'Response to Reviewers'.

We look forward to receiving your revised manuscript.

Kind regards,

Lorenzo Faggioni, M.D., Ph.D.

Academic Editor

PLOS One

Journal Requirements:

Yes, we did.

Yes, we did.

3. Please note that PLOS One has specific guidelines on code sharing for submissions in which author-generated code underpins the findings in the manuscript. In these cases, all author-generated code must be made available without restrictions upon publication of the work. Please review our guidelines at https://journals.plos.org/plosone/s/materials-and-software-sharing#loc-sharing-code and ensure that your code is shared in a way that follows best practice and facilitates reproducibility and reuse.

Yes, we did.

Yes, we did.

Financial disclosure

This research was supported by a grant of the Korean Health Technology R&D Project through the Korea Health Industry Development Institute (KHIDI), funded by the Ministry of Health & Welfare, Republic of Korea (grant number: RS-2022-KH129742). The funders had no role in study design, data collection and analysis, decision to publish, or preparation of the manuscript.

With Best Regards,

Jong-Yeup Kim

Department of Biomedical Informatics, Konyang University College of Medicine, Daejeon, Republic of Korea

jykim@kyuh.ac.kr

none

Yes, we did.

The authors have declared that no competing interests exist.

6. Please update your submission to use the PLOS LaTeX template. The template and more information on our requirements for LaTeX submissions can be found at http://journals.plos.org/plosone/s/latex.

Yes, we did.

7. We note that Figure(s) 1 and 2 in your submission contain copyrighted images. All PLOS content is published under the Creative Commons Attribution License (CC BY 4.0), which means that the manuscript, images, and Supporting Information files will be freely available online, and any third party is permitted to access, download, copy, distribute, and use these materials in any way, even commercially, with proper attribution. For more information, see our copyright guidelines: http://journals.plos.org/plosone/s/licenses-and-copyright.

a. You may seek permission from the original copyright holder of Figure(s) 1 and 2 to publish the content specifically under the CC BY 4.0 license.

Thank you for raising this important point regarding Figures 1 and 2.

We confirm that these figures were created by one of the authors using anonymized CT images obtained from patients enrolled in this study.

The images were edited and annotated solely by the authors and do not contain any copyrighted material from external sources.

As the authors are the original creators and copyright holders of these figures, we confirm that we grant permission for publication under the CC BY 4.0 license.

8. Please remove all personal information, ensure that the data shared are in accordance with participant consent, and re-upload a fully anonymized data set.

Yes, we did.

Yes, we did.

Yes, we did.

Reviewers' comments:

Reviewer's Responses to Questions

Comments to the Author

1. Is the manuscript technically sound, and do the data support the conclusions?

Reviewer #1: No

Reviewer #2: Yes

2. Has the statistical analysis been performed appropriately and rigorously?

Reviewer #1: Yes

Reviewer #2: Yes

3. Have the authors made all data underlying the findings in their manuscript fully available?

Reviewer #1: No

Reviewer #2: Yes

4. Is the manuscript presented in an intelligible fashion and written in standard English?

Reviewer #1: Yes

Reviewer #2: Yes

5. Review Comments to the Author

Reviewer #1: The study addresses an important clinical task; however, several methodological issues need clarification to ensure the soundness of the findings.

1. Title and aim: “predict” vs “measure”.

The manuscript repeatedly refers to “predicting tumour size” and “diagnostic performance”, but the study actually evaluates measurement accuracy and reproducibility of CT-based tumour size compared with pathology and radiologists. Since the reported ICCs show similar agreement between MONCAD LCT and routine reports, the emphasis on “prediction” seems conceptually misleading. I suggest reframing the aim toward measurement accuracy/consistency or defining more precisely what “predict” means in this context.

We sincerely appreciate your thoughtful feedback. The manuscript has been revised to reflect your suggestion.

2. Description of the deep learning–based system.

The Methods section provides very limited information about MONCAD LCT despite describing it as a deep learning–based CDSS. Key functions such as nodule detection are not mentioned in Methods, although detection rates appear in Results. Likewise, the approach to delineating tumour margins, handling part-solid nodules, and determining the maximum solid-component diameter (2D plane vs 3D volumetric search) is not described. Some concise explanation of these components—within the limits of a commercial product—is necessary to understand how the system achieves its reported reproducibility.

Thank you for your insightful comment regarding the need to clarify the technical functions of the MONCAD LCT system. We have revised the <Methods section> and added a supplementary appendix (S1 table) to provide a more detailed description of the system’s architecture and core functionalities, including nodule detection, segmentation, and tumor size measurement.

3. Extraction and definition of radiology report measurements.

The process by which tumour size values were retrieved from routine radiology reports is insufficiently described, especially for part-solid nodules where both total and solid-component diameters may be documented. Without a clear extraction protocol, it is difficult to interpret whether the “report” measurements used for comparison reflect the original clinical assessment or choices made during retrospective data collection.

Thank you for your insightful comment.

We have clarified the process by which tumor size was extracted from the original radiology reports. Specifically, we now state that in part-solid nodules, the solid component’s maximum diameter was used, following the IASLC 8th edition TNM guidelines. We also confirm that no remeasurement was performed and all values reflect the original clinical interpretation. This clarification has been added to the Methods section of the revised manuscript

4. Use of the term CDSS.

Although MONCAD LCT may be marketed as a CDSS, in this study it supports only a narrow component of NSCLC evaluation (size assessment for T staging). It may be helpful to clarify this limited scope or use a more precise term.

Clarifying these points would significantly strengthen the methodological transparency and interpretation of the work.

Thank you for pointing out the need to clarify the scope of the CDSS function in our study.

We agree that, although MONCAD LCT is marketed as a CDSS, in this study it was applied solely for its tumor size measurement function on preoperative CT, relevant to T-staging in NSCLC. To avoid confusion, we have clarified this point in the revised

---

## [Decision Letter · Decision Letter 1]

23 Feb 2026

Accuracy and Reproducibility of Tumor Size Measurement Using a Deep-Learning–Based CDSS in Resected Lung Cancer

PONE-D-25-39492R1

Dear Dr. Kim,

We’re pleased to inform you that your manuscript has been judged scientifically suitable for publication and will be formally accepted for publication once it meets all outstanding technical requirements.

Kind regards,

Lorenzo Faggioni, M.D., Ph.D.

Academic Editor

PLOS One

Reviewers' comments:

Reviewer's Responses to Questions

**Comments to the Author**

Reviewer #1: All comments have been addressed

Reviewer #2: All comments have been addressed

2. Is the manuscript technically sound, and do the data support the conclusions?

Reviewer #1: Yes

Reviewer #2: Yes

3. Has the statistical analysis been performed appropriately and rigorously?

Reviewer #1: Yes

Reviewer #2: Yes

4. Have the authors made all data underlying the findings in their manuscript fully available?

Reviewer #1: Yes

Reviewer #2: Yes

5. Is the manuscript presented in an intelligible fashion and written in standard English?

Reviewer #1: Yes

Reviewer #2: Yes

Reviewer #1: The authors have addressed all of my concerns adequately. I am satisfied with the revisions made to the manuscript.

Reviewer #2: The requested revisions have been incorporated and corrected. The paper still has some limitation , but now the aim and the results are correct

**Do you want your identity to be public for this peer review?** For information about this choice, including consent withdrawal, please see our Privacy Policy

Reviewer #1: No

Reviewer #2: No

---

## [Editor Report · Acceptance letter]

PONE-D-25-39492R1

PLOS One

Dear Dr. Kim,

I'm pleased to inform you that your manuscript has been deemed suitable for publication in PLOS One. Congratulations! Your manuscript is now being handed over to our production team.

Kind regards,

on behalf of

Dr. Lorenzo Faggioni

Academic Editor

PLOS One